# Measuring “Nudgeability”: Development of a Scale on Susceptibility to Physical Activity Nudges among College Students

**DOI:** 10.3390/bs12090318

**Published:** 2022-09-01

**Authors:** Xinghua Wang, Guandong Song, Xia Wan

**Affiliations:** 1The Physical Education Department, Northeastern University, Shenyang 110819, China; 2School of Humanities and Law, Northeastern University, Shenyang 110169, China; 3Social Sport College, Shenyang Sport University, Shenyang 110115, China

**Keywords:** nudges, susceptibility, physical activity, college students, scale development, behavior change techniques

## Abstract

Background: The current college lifestyle create more opportunities for students to develop unhealthy behaviors, especially physical inactivity. Nudging could be an effective tool to improve physical activity behaviors by changing college settings. One-nudge-fits-all leads to ineffective nudges, so it is necessary to develop a reliable and valid instrument capable of measuring the “nudgeability” of physical activity nudges for college students, which is for a higher level of nudge efficacy. Method: Developing the College Physical Activity Nudges Susceptibility Scale (CPANSS) that integrated the nudge method with the Likert scale, which is the first attempt to measure the susceptibility to nudges directly by a scale. There are four steps for developing CPANSS, including Scale Dimensions, Item Generation, Exploratory Factor Analysis (*n* = 294), and Confirmatory Factor Analysis (*n* = 293) with appropriate procedures. Results: The five-factor 21-item CPANSS with good reliability and validity fitted the data reasonably well. Conclusion: The CPANSS was to provide a new tool for policymakers to design effective nudges in changing and promoting physical activity in college settings, and to provide a method for scholars to promote other healthy behaviors for different target groups.

## 1. Introduction

Insufficient physical activity makes individuals have a higher risk of health issues, particularly cardiovascular disease [1]. However, there are 40–50% of college students who are leading inactive lives [2], and it is a social norm to be inactive on campus, that is students sit during lectures, seminars, libraries, as well as lunch. Apparently, the current campus lifestyle and environment [3] creates more opportunities for college students to develop unhealthy behaviors (e.g., physical inactivity and sedentary) [4].

What college settings could help managers and policymakers to promote physical activity among students? The concept of “nudge” is concerned with answering this question. Nudge is based on changing contexts to bring about significant behavior changes [5], which could be used by any choice architect who is in charge of setting up contexts, in which “contexts” refers to the environment within which we make decisions and respond to cues. Since the development of the “Nudge” concept, both governmental and private organizations have shown interest in the application of nudge methods [6]. The attractiveness of using nudge strategies in an era of budgetary constraint stems from the perceived promise to provide “low-cost, unobtrusive” solutions to societal concerns [7]. Nudges are applied in a wide range of fields and in the health sector most frequently [8]. Recently, it has been prevalent in public health policy to use nudge intervention on health risk-related behaviors, such as physical inactivity, smoking, and alcohol abuse, for disease prevention and health promotion [9]. Most nudge studies are based on only one or two interventions. However, researchers would benefit from bringing together the strong effects on behavior so that studies can go deeper into these nudge methods and, if necessary, review and update our understanding of what these nudges are and where they are most effective. Policymakers also need a framework to assist them with drafting policies and using nudge methods to change human behavior.

The MINDSPACE framework was created by the UK Institute for Government and Behavioral Insights Team to aid in the implementation of nudge methods in public policy and for behavior change [10,11]. MINDSPACE was arranged according to the acronym of nine types of behavior change techniques (BCTs), which are Messenger, Incentives, Norms, Defaults, Salience, Priming, Affect, Commitment and Ego. These nine elements have been demonstrated many times to have significant impacts on behavior change, backed up by extensive research in the fields of social psychology and behavioral economics. Currently, many review studies gathered and classified nudge strategies by the MINDSPACE framework, which are systematized in a framework that can be used as a “checklist” by scholars and policymakers [11].

However, as nudging has progressed, we have begun to understand some of its shortcomings. A nudge may appear to be effective because a population benefited on average, but it may be different at the individual level. According to some findings, one-size-fits-all nudge designs will fall short of achieving population-wide goals and even harm sub-groups of a population, which means the effectiveness of nudges also depends on the types of nudges used [12,13]. For instance, in a nudge study on encouraging rational consumption [14], spendthrifts spent less when nudged, moving them closer to the optimal spending range, whereas tightwads similarly spent less, moving them away from the optimal. Another example is the use of nudging in the canteen [15], which improved staff healthy eating behavior but led to the opposite behavior among students. These findings were attributed to differences in nudging susceptibility, which is the “nudgeability” of different nudge types is different for target groups. When the individuals are not susceptible to this nudge, it can also yield the nudge ineffective. Therefore, a greater susceptibility of target groups to nudges is to show a better “nudgeability” of this nudge type, which is related to a higher level of nudge efficacy.

In the present study, we would like to develop a scale to measure the susceptibility of college students to physical activity nudges in the college setting. The development of the College Physical Activity Nudging Susceptibility Scale (CPANSS) is an integration of nudges with the Likert scale, which classifies nudges based on the MINDSAPCE framework and describes the nudge interventions as items. By doing so, we provide a policy instrument that could contribute to the nudging study on measuring the “nudgeability” of nudges for a target group and determining what kinds of nudges or BCTs are most effective in the specific circumstances.

## 2. Literature Review

Before offering research examples of nudging physical activity behavior change, the current study elaborated on dual process theory and behavioral systems for understanding how nudges influence behavior change.

### 2.1. Theoretical Background

#### 2.1.1. Dual Process Theory

Dual processing theory is now largely acknowledged as the most common explanation for the cognitive processes that underpin human decision-making, especially for understanding physical activity behaviors [16,17]. It is assumed by both psychologists and neuroscientists that brain functioning is based on two types of processes to act a behavior: “System 1” (automatic system) processes as being fast, which are described as automatic, effortless, fast, unconscious, and affective; “System 2” (reflective system) processes are slow, which are described as reflective, controversial, and effective [18,19]. This dual process model has been bolstered by evidence of different brain regions for automatic information processing [20]. Much of our daily behavior is determined by “System 1” [21,22]. Moreover, human behaviors are influenced by irrational social and environmental factors in real life [23].

Usually, traditional public policy initiatives have depended on the reflecting mind (System 2) to change behavior, partly because mainstream economic theories and the rational choice framework, in general, are so dominant [5]. Traditional public policy constantly employs education campaigns and various types of incentives to alter people’s cognitive appraisals of the costs and advantages of certain decisions. Unfortunately, this strategy leaves a significant percentage of the variation and bias in behavior to be explained [24]. For instance, David Laibson, a Harvard economist, frequently uses the example of exercise to highlight our profoundly ingrained procrastination. We all know about the benefit to our health of doing exercise, but exercising today entails immediate sweat and effort, with the advantages accruing over time; as a result, we are less likely to exercise as often as we should [25].

Because they rely on modifying the environment in which the individual behaves without necessarily affecting the underlying cognitions, studies concentrate on the generally context-based drivers (System 1) of behavior to explain cognitive biases and constrained rational behavior [26]. As a result, the dual process model is suggested as a theoretical foundation for nudge theory [7].

#### 2.1.2. Behavioral Systems Controlling Behavior

Human decision-making can be simplified using dual process models of behavior. However, there are worries that these models’ basic dichotomy may fail to completely portray the complexity of individual decision-making. The dual process model is used by Ivo Vlaev’s team [27] to give a more detailed description of how automatic processes govern behavior and how nudges influence impulsive behaviors. Based on recent developments in cognitive neuroscience and behavioral economics, this new view is presented [28,29]. There are three primary brain systems (Goal-Directed System, Impulsive System, and Habit System) in the self-regulatory processes involved in behavioral change, each of which can generate psychological processes, including thinking, planning, drives, emotions, and mental and motor habits that drive behavior [30]. Vlaev and Dolan [31] provide an overview of those brain areas as well as an assessment of the evidence.

Impulsive system and habit system are for automated processing, according to recent integrative theories and reviews [32]. The impulsive system is built on effective reactions that have evolved, which are generated in response to certain environmental cues (e.g., people, food, money) and lead to a decision or behavior [33,34]. The habit system is based on learning to assign values to a variety of acts according to the rewards and penalties experienced as a result of performing those actions through frequent repetition in a stable environment [35] with evidence of changes in an individual’s neurobiology [33]. As a result, contextual or environmental cues can subconsciously evoke a desire or trigger a habitual response [7].

### 2.2. Nudges in Physical Activity

Since the publication of Thaler and Sunstein’s landmark book Nudge in 2008, the use of behavioral insights and nudges to change behaviors has gained favor among public and commercial institutions. Despite the fact that physical activity has been proved to be useful in avoiding numerous chronic diseases [36], population-wide physical activity is still low [37]. Population-based techniques that explicitly consider context and environmental elements in decision-making could be a viable alternative for increasing daily physical activity. However, there are few choice architecture interventions in the general population that target physical activity.

Currently, physical activity research always focuses on certain target groups [38], specific settings [39], disease prevention [40], and technology use [41]. Soler et al. [42] released articles in 2010 that looked at the point-of-choice prompts. Zimmerman and his colleagues [43] looked at behavioral economics with the goal of encouraging people to be more active. He advised going beyond the default option, such as pre-selected possibilities, and instead focusing on “anchors”, which are reference points (e.g., norms, frames, habits) that influence later judgments, to observe how people interact with their environment and how their preferences are influenced. However, nothing beyond the use of point-of-choice nudges is known about population-level interventions for physical activity promotion. Some scoping study [44,45] tries to fill that vacuum by Sunstein and Thaler [5], providing an overview of the scope of nudge interventions to promote physical activity in the general population and assessing the various approaches in terms of the class and kind of intervention utilized. Moreover, these scoping reviews research classified nudges in physical activity according to the MINDSAPCE framework, for instance, Sarah, Frauke, and Chiara [44] found that nudges interventions were applied in 26 of the 256 articles to promote physical activity and reduce sedentary behavior in the workplace, and they also indicated that the frequency of nudging techniques to promote PA in MINDSPACE factors was “messenger” (*n* = 4), “incentives” (*n* = 3), “norms” (*n* = 3), “defaults” (*n* = 1), “salience” (*n* = 15) “priming” (*n* = 14), “affect” (*n* = 4), “commitment (*n* = 1)” and “ego” (*n* = 1). “Priming”, “salience”, and “messenger” were the three most widely employed nudging approaches to enhance PA.

## 3. Methods

Prior research on the use of nudge intervention to change behaviors has many limitations. Usually, research with a nudge or behavioral insight approach is typically characterized by a focus on: (a) one location; (b) a small sample; (c) a limited set of tools; (d) single actor type; (e) single response of behaviors; and (f) experimental methods [46,47,48]. It is suggested to do more to integrate the nudge and traditional psychology methods to expand the range of measurements and develop a new instrument with a nudge approach to change human behaviors [31]. In an effort to develop a wide range of tools with the nudge approach, we would like to develop a scale to measure the susceptibility to nudge influence, which is an integration with nudges and the Likert scale. This method not only expands the range of measurement but also adds the function of the instrument. The scale classified nudges based on the MINDSAPCE framework and described the nudge interventions as items that could help policymakers to measure the “nudgeability” of different nudge types for a target group and to make sure the effectiveness of nudge strategies applied in a public policy.

In the current study, we applied a mixed methods design. The design included: (a) a qualitative component (Phase 1) to classify the dimensions of physical activity nudges based on the MINDSPACE framework, and to describe the nudges based on the findings of nudge interventions studies and the scoping review studies in nudge, then to preliminary develop the College Physical Activity Nudging Susceptibility Scale (CPANSS); and (b) a quantitative component (Phase 2) to examine the qualitative results on a large sample, to provide evidence for construct reliability as well as convergent, discriminant, and concurrent validity of the CPANSS.

### 3.1. Phrase 1

The purpose of Phase 1 was to ensure the dimensions (types) of physical activity nudges and to generate items based on the findings of the literature review, which is to explore a preliminary version of the College Physical Activity Nudging Susceptibility Scale (CPANSS).

#### 3.1.1. The Dimensions of Physical Activity Nudges

The MINDSPACE framework provides researchers with a valid tool for classifying nudges. However, there are some types of nudges that are frequently combined in nudge intervention studies. For instance, it has been discovered that combining salience, priming, and affect nudges to change behaviors is particularly effective [49]. Similar results were also obtained when the commitment was combined with ego and applied a combination nudge of messenger and norms [46]. Therefore, the dimensions of physical activity nudges are as follows: (i) messenger and norms, (ii) incentives, (iii) default, (iv) salience, priming, and affect, and (v) commitment and ego. The descriptions of the five dimensions are shown in Table 1.

#### 3.1.2. Item Generation

Based on the five dimensions mentioned above, scale items were generated depending on the intervention study applied to each type of nudges. We described and classified nudge interventions from 29 research studies as nudging situations in the college setting, as shown in Table 2. Most of these intervention studies were applied in PA, for example, ‘7. When a college asks students to use sports facilities more frequently in official emails.’ it was adapted from two interventions to encourage stair use via staff emails [48,49]. In addition, ‘16. When stairs located next to escalators were transformed into a working piano, with every step playing a note to the public.’ is a famous social experiment called “Musical Stairs” developed by the Fun Theory team in 2009 [50] of Volkswagen. Although some of the researches were applied in other fields, for instance, ‘4. When the school informed you that 70% of the students ran on the playground today.’ was adapted from a successful intervention named ‘Most of Us Wear Seatbelts Campaign’ to boost the number of individuals wearing seatbelts, an aggressive social norms media campaign was initiated to notify communities that 85% of responders used a seatbelt [51].

Moreover, further elaboration is required about the number of items in the Default factor. Most methodologists endorse that each factor should include a minimum of three variables; however, at least four variables per dimension are recommended [52]. The Default factor is a two-item factor in this study, as research using default nudge is currently quite rare. We only found two related nudge intervention strategies that could be applied to physical activity behavior change. According to certain studies, 15.8% of journal articles in their new scales had two-items factors [53]. Two-items scales are only used if the items are highly correlated (i.e., r < 0.70) [54]. In terms of CPANSS, it is reasonable because D1 and D2 are substantially connected (i.e., r = 0.752 < 0.70).

It should be clarified that all of the original items were created in Chinese since the participants of this study are college students from China. For this manuscript, the items were translated into English. Bilingual research assistants who had graduated from both Chinese- and English-language programs conducted the translation and back-translation to match the original items as recommended by Bracken and Barona [55]. In terms of the translation procedure, items were initially translated into English by one research assistant and back-translated into Chinese by another. The principle of translation was for conceptual rather than literal meaning. For the appropriateness and equivalence between the Chinese and English versions, the differences found in the back-translation were discussed and modified by the translator, back-translator, and the monolingual Chinese-speaking member of the study team.

### 3.2. Phrase 2

#### 3.2.1. Measurement

Phase 2 utilized a quantitative approach to corroborate the scale constructs obtained from Phase 1. A measurement scale, the College Physical Activity Nudging Susceptibility Scale (CPANSS), was subsequently developed based on findings from Phase 1 through exploratory factor analysis (EFA) and was verified via confirmatory factor analysis (CFA).

The items were measured on a 5-point Likert-type scale (1 = not at all susceptible; 2 = not very susceptible; 3 = moderately susceptible, 4 = very susceptible; 5 = extremely susceptible). The items were preceded by the following statement: ‘Circle the number that best describes your susceptibilities to these physical activity nudges in situations’. Despite the CPANSS items, the survey also included sociodemographic variables.

#### 3.2.2. Data Collection and Procedures

Data were gathered over one week by using Wenjuanxing, which is the largest professional platform for online surveys in China, with more than 2.6 million sample sources [80]. Questionnaires were sent to 622 college students from a university in northeastern China with 587 respondents (94.4%), and the sample size was larger than ten times the number of analyzed items (*n* = 24), meeting the requirements for sample size. It was completely voluntary for all participants in this study. Prior to filling in the questionnaire, all participants had obtained informed consent and were introduced in detail to the objectives and purposes of the study. We made it clear to all participants that the data would only be used for academic study.

Of the participants, there are 301 males (51.3%) and 286 females (48.7%). The ratio of the sample in different grades is as follows: grade 1 (31.7%), grade 2 (27.3%), grade 3 (27.0%), and grade 4 (14.0%). Moreover, samples with three types of majors: social science majors (*n* = 223), nature science majors (*n* = 301), and sports majors (*n* = 63).

#### 3.2.3. Data Analyses

We divided the quantitative sample into two halves at random, each with the same sample size. The Exploratory Factor Analysis was used on the first set of the sample (*n* = 294) to identify constructs, and the Confirmatory Factor Analysis was used on the second set (*n* = 293). The comparability of the EFA sample and CFA sample in demographic variables was verified by using the chi-square test. As shown in Table 3, demographics of gender (χ^2^ = 2.331, *p* = 0.127 > 0.05), major (χ^2^ = 5.469, *p* = 0.065 > 0.05), and grade (χ^2^ = 6.464, *p* = 0.091 > 0.05) were not significantly associated with the type of sample, indicating that the demographics of the EFA sample and the CFA sample were comparable.

Calculating descriptive statistics, chi-square tests, reliability, and conducting the EFA were all done using the SPSS 26.0 program. The CFA analyses with AMOS version 25 procedures. χ^2^/df < 3, CFI > 0.90, NNFI > 0.90, TLI > 0.90, IFI > 0.90, and RMSEA < 0.08 were used as goodness of fit (GOF) indices [81]. To verify the reliability, convergent validity, and discriminant validity of the CPANSS, Cronbach’s alpha (α), construct reliability (CR), and average variance extracted (AVE) values were assessed [82].

## 4. Results

### 4.1. The EFA

An Exploratory Factor Analysis was performed (*n* = 294) to identify a simple structure for the CPANSS. For the physical activity nudge factors, the Kaiser-Meyer-Olkin measure of sampling adequacy value was 0.888 (>0.70), which exceeded the cut-off value of 0.40, implying that the level of common variance was justified [81]. The Bartlett’s Test of Sphericity (*p* < 0.001) with 3292.922, indicating that at least some of the variables in the correlation matrix had significant correlations, which means an EFA was determined to be appropriate for the sample.

Three items were discarded. Two items (‘6. When you see that all the students in the gym are exercising.’; ‘20. When the school awards honorary certificates to students who are active.’) with loading less than 0.50 were considered inadequate indicators for that factor [83,84]. Moreover, these two items were eliminated due to their inappropriate contents. Respondents may consider the situation described by Item 6 that students who exercise at the gym are already fitness enthusiasts, which could not influence them much. Item 20 is the only Commitment and Ego nudge on honorary and the sports honorary certificate for respondents with insufficient attraction because of the perception of sports. Another item (‘18. When smelling fresh and comfortable outdoors.’) described a nudge situation of SPA but loading on MN at 0.676. It was removed due to its content being entirely unrelated to MN.

Consequently, the five physical activity nudge factors were discovered statistically and formally labeled as messenger and norms (6 items), incentives (3 items), default (2 items), priming, salience, and affect (5 items), as well as commitment and ego (5 items).

Five dimensions with 21 items satisfying the retention criterion, accounting for 65.55% of the variance among the variables. The nine elements of the MINDSPACE framework were considered correlated, so we conducted the EFA with direct oblimin rotation (a kind of oblique rotation) in the present study. It is recommended to use oblique rotation because it more accurately represents most models, and it allows factors to correlate [85,86]. The results of the rotated pattern matrix from direct oblimin rotation were presented in Table 4.

### 4.2. The CFA

To verify the model of the suggested scale obtained from the process of EFA, a CFA should be performed on a distinct sample [54,87,88]. We utilized AMOS version 25 to perform the CFA analysis (*n* = 293) for the CPANSS variables. Confirmatory Factor Analysis (CFA) is a statistical examination of survey data that is used to see if the relationship between a factor and the associated observed variable matches the researcher’s theoretical relationship.

The five-factor 21-item CPANSS fitted the data reasonably well, according to the goodness of fit (GOF) measures [81,89], which are shown in Table 5. The χ^2^ value (349.198) was significant at *p* < 0.001, and the normed chi-square (χ2/df = 1.95) was lower than 3 (benchmark value), indicating that the fit was satisfactory. The Root Mean Square Error of Approximation (RMSEA) statistic showed a reasonable fit for the CPANSS (RMSEA = 0.057 < 0.08). The CFI value of 0.957, IFI value of 0.957, and NNFI (TLI) value of 0.949 were all within the ideal range (>0.90), indicating that the fit was good. Standardized factor loadings of all items were beyond the ideal value of 0.70 [81], and the structure of the CPANSS model was shown in Figure 1.

As shown in Figure 1, the covariance between SPA and MN is high at 0.72 but lower than 0.85, which means the discriminant validity between this pair of factors is acceptable, but a second order construct could be considered for a more parsimonious model. It has been emphasized that the ultimate justification for using higher-order constructs is the theory, which means that higher-order constructs should not be employed if they do not theoretically make sense [53,81]. The dimensions of CPANSS are the types of nudges; however, at present, there are no nudge strategies that combine both MN and SPA, and no study or theory is showing that there is a latent variable or a concept that could explain both MN and SPA. Therefore, a second-order construct will not be considered for now in the present study.

As indicated in Table 6, Cronbach’s alpha of each dimension was higher than 0.7 (α > 0.70), indicating a good consistency or stability of the scale’s measurement results. Standardized factor loadings (λ> 0.70), construct reliability (CR > 0.70), and average variance extracted (AVE > 0.50) values were used to assess the scale’s good convergent validity. Moreover, as shown in Table 7, the scale has good discriminant validity because the square root of the AVE value of each variable is bigger than the correlation coefficients between variables.

## 5. Discussion

The purpose of the present study was to develop a reliable and valid instrument capable of measuring the susceptibility to nudging situations of college students. The reasons for developing a scale that integrated the nudge method with the Likert scale are due to the limitations of previous nudging methods, such as the narrow measuring range, and also because the situation of one-nudge-fits-all leads to ineffective nudges. This method provided by us would contribute to measuring the “nudgeability” of different types of nudges for particular populations, which is a good attempt to help policymakers to design effective nudges.

In terms of the process of developing this scale, we followed appropriate procedures in scale development. We made some combinations of nudge types in the MINDSPACE framework as dimensions of scale based on the findings that these types of nudges are frequently combined and applied in nudge intervention with higher effectiveness. We classified physical activity nudges as (i) messenger and norms, (ii) incentives, (iii) default, (iv) salience, priming, and affect, and (v) commitment and ego, which are similar to a previous review [90].

Although it is not the first attempt to develop a scale on nudges, it is the first scale to measure the susceptibility to nudges. There are two scales measuring the acceptability of nudging in sustainable eating behaviors developed by Laurens’s team [91] and Nørn-berg’s team [92]. Acceptable means satisfactory and able to be agreed to or approved of; susceptible means easily influenced by something. Therefore, acceptability indicates a degree of approval for the nudge strategies, whereas susceptibility is more about how individuals are influenced by nudges in situations. Sometimes, it is possible for people to approve of the nudge strategies but not be affected by them. Therefore, measuring susceptibility to nudging is a much more appropriate and direct indicator of nudging effectiveness than acceptability.

Moreover, it is also a creative attempt to describe nudges as situation items on a scale. The HABITS Lab of the University of Maryland Baltimore County has developed some situation temptation scales with 9, 12, or 20 items to measure how tempted an individual is to engage in a variety of different unhealthy behaviors such as smoking, drug addiction, alcoholism, and eating behaviors. In their situation temptation scales, situation items were classified into different subscales (dimensions). For example, Alcohol Situation Temptation Scale classified situations into four dimensions which are Negative Affect, Social/Positive, Physical and Fatigue, and Cravings and Urges. Respondents could measure their temptation of each subscale by summing item scores for each subscale and dividing by the number of items. This kind of scale would help individuals to understand and examine the situations in which they are more vulnerable to temptation in unhealthy behaviors and help them to reduce or stop engaging in those behaviors. However, there is no situation scale for promoting healthy behavior yet. CPANSS is a good attempt to help inactive people understand the situations in which they are more likely to be active.

## 6. Conclusions

In summary, the present study developed a scale to measure the susceptibility of college students to physical activity nudges (i.e., CPANSS). This scale is a self-report measure scale, which makes the first attempt to measure the “nudgeability” of nudges situations by a scale method. For developing CPANSS, there are four steps with appropriate procedures in scale development. Therefore, CPANSS could be a reliable and valid instrument for researchers and policymakers to change and promote physical activity in college settings. Moreover, developing a scale that integrated the nudge intervention with the Likert scale could provide a method for measuring the “nudgeability” of nudges for a target group, which could also be applied in other settings and behavior changes.

## 7. Limitations and Future Study

The current study has some limitations that should be mentioned. This scale is developed based on the sample of Chinese college students, so cross-sample and cross-cultural studies are still needed to determine its generalizability, which in turn can be used for studies with broader samples. Another limitation in the present study is the dimension Default of CPANSS. In the process of item generation of Default, we only found two related nudges that have been applied in physical activity behavior change. Although D1 and D2 with high correlations to make the Default dimension reasonable, it is still suggested to generate more items when more default nudges interventions are applied in PA in the future.

It should be encouraged for researchers to explore the interrelationship between MN and SPA due to the finding of a high covariance between them, which will help a better understanding of the MINDSPACE framework and improve the classification of nudge types. Furthermore, exploring the reasons for the correlation between the nudge elements could help us to design a more effective nudge intervention in the future.

Because of the urgency of college students’ inactivity issues, the present study focused on college students. For future study, researchers could explore the context of scale. However, some items of CPANSS are just for the campus, such as ‘When your school establishes a “sports scholarship” to reward students who have good performance in sports events.’, which is easy to modify in the context of the workplace such as ‘When your department creates a “sports bonus” to reward staffs who excel in sports activities.’. CPANSS could also be explored in other educational systems, such as primary school and middle school; be explored in other regions, such as Europe and Africa; be explored in other contexts, such as community and city parks; be explored in other target groups, such as older people, E-sports players, and citizens; be explored in other behaviors, such as environmental behavior, prosocial behavior, transportation, and eating or drinking habits.

## Figures and Tables

**Figure 1 behavsci-12-00318-f001:**
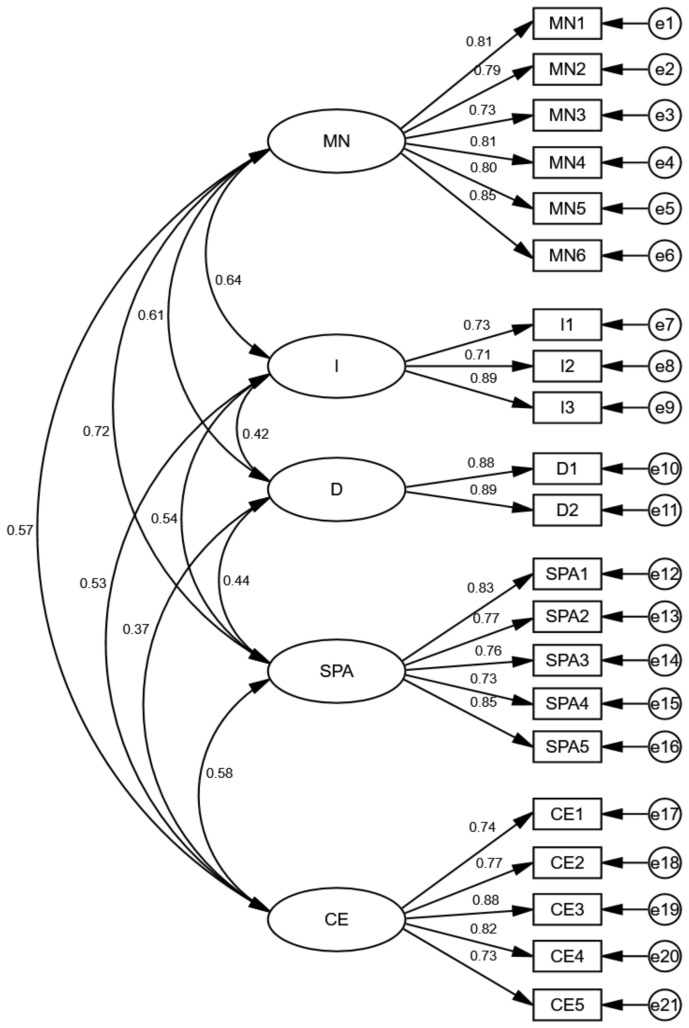
The CFA result of the structure of the CPANSS. Note. MN = Messenger and Norms; I = Incentives; D = Default; SPA = Salience, Priming, and Affect; CE = Commitment and Ego. Path coefficients presented in standardized. Note that each item has an error term with it (labeled e1-e21).

**Table 1 behavsci-12-00318-t001:** The descriptions of dimensions.

Dimensions	Description
Messenger and Norms	The person or organization delivering the information and to establish a norm.
Incentives	Penalties to discourage a negative choice or rewards to encourage a positive option.
Defaults	A specific choice is ‘preset,’ making it the simplest option.
Salience, Priming and Affect	Innovative, individualized, and compelling explanations and examples are utilized to draw attention to a certain decision, even subconsciously.
Commitments and Ego	Making a public promise or a commitment in order to enhance one’s desire to feel good about oneself.

**Table 2 behavsci-12-00318-t002:** Nudging items generation.

Categories	Nudging Situation Items	Source
Messenger & Norms	1. When your friends invite you to participate in sports activities. 2.When your family participates in daily physical activity. 3. When students on campus are always using a shared bike. 4. When the school informs you that 70% of the students ran on the playground today. 5. When your friend tells you he/she is working out. 6. When you see that all the students in the gym are exercising. 7. When a college asks students to use sports facilities more frequently in official emails.	[12,48,49,51,56,57,58,59,60,61]
Incentives	8. When your school establishes a “sports scholarship” to reward students who have good performance in sports events. 9. When the amount of physical activity as installments to buy expensive goods. 10.When you have a chance to win a prize if you participate in physical activity.	[57,58,62,63,64]
Default	11. When the number of steps today is not yet 10,000 steps. 12. When the workout software plan’s calorie goal has not yet been met.	[58,65].
Salience, Priming & Affect	13. When you’re telling others about a fantastic sporting event you experienced. 14.When viewing pictures of the consequences of sedentary and inactivity in the library. 15. When you find reports clearly pointing out the risk of sickness in college student groups who do not exercise. 16. When stairs located next to escalators were transformed into a working piano, with every step playing a note to the public. 17. When the number of calories burned is indicated on each step of stairs by the elevator. 18. When smelling fresh and comfortable outdoors.	[48,50,66,67,68,69,70,71,72,73,74,75]
Commitment & Ego	19. When you promised a friend or family member that you would exercise every day. 20. When the school awards honorary certificates to students who are active. 21. When making a promise with others that you could meet your physical activity goal. 22. When you make a deal with someone, you will be rewarded if you work out for a specific amount of time. 23. You will get recognition from friends and family when getting the set workout goal. 24. When you promised to meet a specific workout goal as part of a contract.	[68,69,76,77,78,79]

**Table 3 behavsci-12-00318-t003:** Chi-square results for demographic variables.

Characteristics	EFA *n* = 294	CFA *n* = 293	χ^2^	*p*
Gender			2.331	0.127
Male	160	141		
Female	134	152		
Major			5.469	0.065
Social science	99	124		
Nature science	158	143		
Physical education	37	26		
Grade			6.464	0.091
Grade 1	85	101		
Grade 2	91	69		
Grade 3	83	76		
Grade 4	35	47		

**Table 4 behavsci-12-00318-t004:** Factor pattern matrix for the CPANSS: EFA with principal-components extraction and direct oblimin rotation (*n* = 294).

Construct	Items	F1	F2	F3	F4	F5
Messenger and Norms	MN1	0.700				
	MN2	0.705				
	MN3	0.658				
	MN4	0.746				
	MN5	0.634				
	MN6	0.796				
Incentives	I1		0.833			
	I2		0.828			
	I3		0.758			
Default	D1			0.890		
	D2			0.867		
Salience, Priming, and Affect	SPA1				0.768	
	SPA2				0.611	
	SPA3				0.704	
	SPA4				0.690	
	SPA5				0.775	
Commitment and Ego	CE1					0.801
	CE2					0.721
	CE3					0.735
	CE4					0.740
	CE5					0.829

**Table 5 behavsci-12-00318-t005:** The goodness of fit (GOF) measures (*n* = 293).

Common Indices	χ2	df	χ2/df	RMSEA	CFI	IFI	NNFI	TLI
Judgment criteria	-	-	<3	<0.08	>0.90	>0.90	>0.90	>0.90
Target Value	349.198	179	1.95	0.057	0.957	0.957	0.949	0.949

**Table 6 behavsci-12-00318-t006:** Analysis results of reliability and convergent validity for the finalized CPANSS (*n* = 293).

Construct	Item	S.E.	t	*p*	Std.	α	CR	AVE
Messenger and Norms (6 items)	MN1	-	-	-	0.814	0.914	0.914	0.640
	MN2	0.061	15.433	0.000	0.792			
	MN3	0.058	13.93	0.000	0.734			
	MN4	0.058	15.823	0.000	0.806			
	MN5	0.055	15.598	0.000	0.798			
	MN6	0.064	17.031	0.000	0.849			
Incentives (3 items)	I1	-	-	-	0.730	0.816	0.823	0.610
	I2	0.07	11.358	0.000	0.712			
	I3	0.092	12.982	0.000	0.888			
Default (2 items)	D1	-	-	-	0.882	0.880	0.787	0.881
	D2	0.074	13.696	0.000	0.892			
Salience, Priming and Affect (5 items)	SPA1	-	-	-	0.826	0.891	0.891	0.622
	SPA2	0.054	14.808	0.000	0.770			
	SPA3	0.055	14.665	0.000	0.765			
	SPA4	0.061	13.686	0.000	0.726			
	SPA5	0.063	16.992	0.000	0.851			
Commitment and Ego (5 items)	CE1	-	-	-	0.743	0.892	0.892	0.626
	CE2	0.078	13.13	0.000	0.771			
	CE3	0.079	15.095	0.000	0.883			
	CE4	0.082	14.008	0.000	0.819			
	CE5	0.073	12.372	0.000	0.730			

**Table 7 behavsci-12-00318-t007:** Analysis results of discriminant validity for the finalized CPANSS (*n* = 293).

	MN	I	D	SPA	CE
**MN**	**0.800**				
**I**	0.547	**0.781**			
**D**	0.541	0.351	**0.887**		
**SPA**	0.638	0.465	0.391	**0.789**	
**CE**	0.512	0.463	0.331	0.513	**0.791**

Note: The items on the diagonal in bold represent the square roots of the AVE.

## Data Availability

The datasets used and analyzed during the current study are available from the corresponding author on reasonable request.

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
