# Peer review of "Measuring “Nudgeability”: Development of a Scale on Susceptibility to Physical Activity Nudges among College Students"

_behavsci, 2022, doi:10.3390/bs12090318_

Round 1

Reviewer 1 Report

Congratulations to the Authors of an interesting text and research results.

The article presents the results of research on the issue of the "nudgeability" and ist measuring. The Authors indicate that the nudging strategy can be a good way to increase the physical activity of college students.

In terms of methodology, the text was very well prepared.

I would like the Authors to extend only the conclusions from the research. Can the applied research methodology be used in other educational systems? Can the research results be used, for example, in European education systems?

The text does not require major editorial corrections.

After completing the conclusions, I recommend the text for publication.

Author Response

Dear reviewer:

Thank you for reviewing our manuscript and for your confirmation of the merits and potential of our study. With consideration of your suggestion, we have improved our work. All the changes are highlighted in blue text in this revision. Your comments provided us (certainly readers of this article, as well) with a new direction for continued research. Our point-by-point response to the comments made by the reviewers is also included in the attachment.

Reviewer 2 Report

This manuscript is centred around the development of a scale to measure the susceptibility of college students to physical activity nudges in a college setting. The impetus for developing the CPANSS is well-presented; a search in Google Scholar did not yield any similar measures. Please consider the following comments:

1. The manuscript needs to be proofread. There are quite a few grammatical errors.

2. I could have missed this, but was the CPANSS administered in Chinese? If it was, the authors should consider specifying that, and for the purposes of this manuscript, the items were translated into English. Efforts to establish equivalence between the Chinese and English versions of the CPANSS should be documented within the manuscript.

3. line 247 - no need to specify "there are more than half of participants were male ...". Just state N and percent.

4. line 252 - please specify comparability of both split-half samples. Most studies do this via a simple chi-sq reporting. This is important as it convinces readers that the EFA sample was comparable to the CFA sample in terms of demographics (e.g., gender, majors/grades).

5. Table 2 - it is untenable to have only 2 items during the item generation stage for a subscale (i.e., default) based on best practices. A bare minimum of three, preferably four items might be more appropriate. Do explain why the "default" subscale comprised only 2 items during item generation - was there a reason to be parsimonious?

6. Lines 270 to 272 - it's creditworthy that you applied the principle of cross loading and the loading greater than 0.5 to identify misfitting items. Nonetheless, there are reputable sources (e.g., Kline) that specify that item loadings should be at least 0.3. In this regard, you might not have needed to discard the items. If you still prefer not to consider the items that were discard, could you also include an explanation from the "content appropriateness" perspective? Why might these items misfit? Could it have been perceived by the respondents in an unexpected way? Solely relying on the 0.5 cutoff and cross loading appears inadequate.

7. Lines 279 and 281 - did you use varimax or oblimin rotation? Please also explain why you used varimax (or oblimin). It's more common for studies within the social sciences to use oblique rotation. In fact, you should clarify why you used varimax (or oblimin) when you allowed the factors to correlate in the CFA (Figure 1). Varimax doesn't allow factors to correlate.

8. Line 296 - please specify referencing that states that standardised loadings should be greater than 0.7.

9. The covariance between subcale SPA and MN is 0.72.  You could consider a second order CFA.

Author Response

Dear reviewer:

The authors want to thank the reviewers for the questions and comments, which are very helpful in improving the quality of our paper. Considering all the comments, we have made substantial changes in this revision. All the changes are highlighted in blue text. Our point-by-point response to the comments made by the reviewers is also included in the attachment.

Round 2

Reviewer 2 Report

Thank you for taking time to address my comments. The manuscript is well-presented given the edits made.